# Steering ecological-evolutionary dynamics to improve artificial selection of microbial communities

Li Xie[1✉] & Wenying Shou [2✉]

Microbial communities often perform important functions that depend on inter-species interactions. To improve community function via artificial selection, one can repeatedly grow many communities to allow mutations to arise, and "reproduce" the highest-functioning communities by partitioning each into multiple offspring communities for the next cycle. Since improvement is often unimpressive in experiments, we study how to design effective selection strategies in silico. Specifically, we simulate community selection to improve a function that requires two species. With a "community function landscape", we visualize how community function depends on species and genotype compositions. Due to ecological interactions that promote species coexistence, the evolutionary trajectory of communities is restricted to a path on the landscape. This restriction can generate counter-intuitive evolutionary dynamics, prevent the attainment of maximal function, and importantly, hinder selection by trapping communities in locations of low community function heritability. We devise experimentally-implementable manipulations to shift the path to higher heritability, which speeds up community function improvement even when landscapes are high dimensional or unknown. Video walkthroughs: https://go.nature.com/3GWwS6j; https://online.kitp.ucsb.edu/online/ecoevo21/shou2/.

[1] Basic Sciences Division, Fred Hutchinson Cancer Research Center, Seattle, United States. [2] Centre for Life's Origins and Evolution, Department of Genetics, Evolution and Environment, University College London, London, United Kingdom. ✉email: xie-li@outlook.com; wenying.shou@gmail.com

Multispecies microbial communities often display *community functions*—biochemical activities not achievable by any member species alone. For example, a community of *Desulfovibrio vulgaris* and *Methanococcus maripaludis*, but not either species alone, converts lactate to methane in the absence of sulfate[1]. Community function arises from "interactions" where one community member influences the physiology of other community members. Interactions are typically complex and difficult to characterize, making it challenging to rationally design communities[2,3]. In a different approach, one could mutagenize individual community members, assemble them at various ratios, and screen the resultant communities for high community function. However, this requires community members to be culturable, and the number of combinatorial possibilities increases rapidly with the number of species and genotypes. In addition, such assembled communities might be vulnerable to ecological invasion[4].

Alternatively, community function may be improved by artificial selection (directed evolution; Fig. 1a)[5–7]. During each selection cycle, newly assembled Newborn communities ("Newborns") grow into Adult communities ("Adults") over a period of "maturation" time set by the experimentalist. During community maturation, community members can proliferate and mutate. At the end of community maturation, Adults expressing the highest community function are chosen to "reproduce" where each is randomly partitioned into multiple Newborns for the next selection cycle. Artificial community selection, if successful, can improve useful community functions such as fighting pathogens[8], producing drugs[9], or degrading wastes[10] without detailed knowledge of the underlying mechanisms.

Theoretical work predicts that artificial selection of communities can succeed, at least under certain conditions[4,11–16]. Experimental work on community selection often yielded variable outcomes[17–28], and some studies were not conclusive due to the lack of a "no selection" control. In some cases, communities indeed responded to selection, presumably driven by changes in species composition[22–25] and/or evolution[17,18]. In other cases, selecting for high-function communities yielded similar outcomes as selecting for low-function or random communities[19,26–28], and community function could even decline despite selection[20,25]. For example, selecting marine microbial communities for enhanced chitin-degradation activity was ineffective, unless community maturation time was progressively adjusted to prevent undesirable species from taking over[25].

Successful community selection requires three elements: variation in community function, preferential survival of high-functioning communities, and heritability of community function[29]. Preferential survival of high-functioning communities is enabled by intercommunity selection. Variation and heritability of community function can be understood in terms of variation

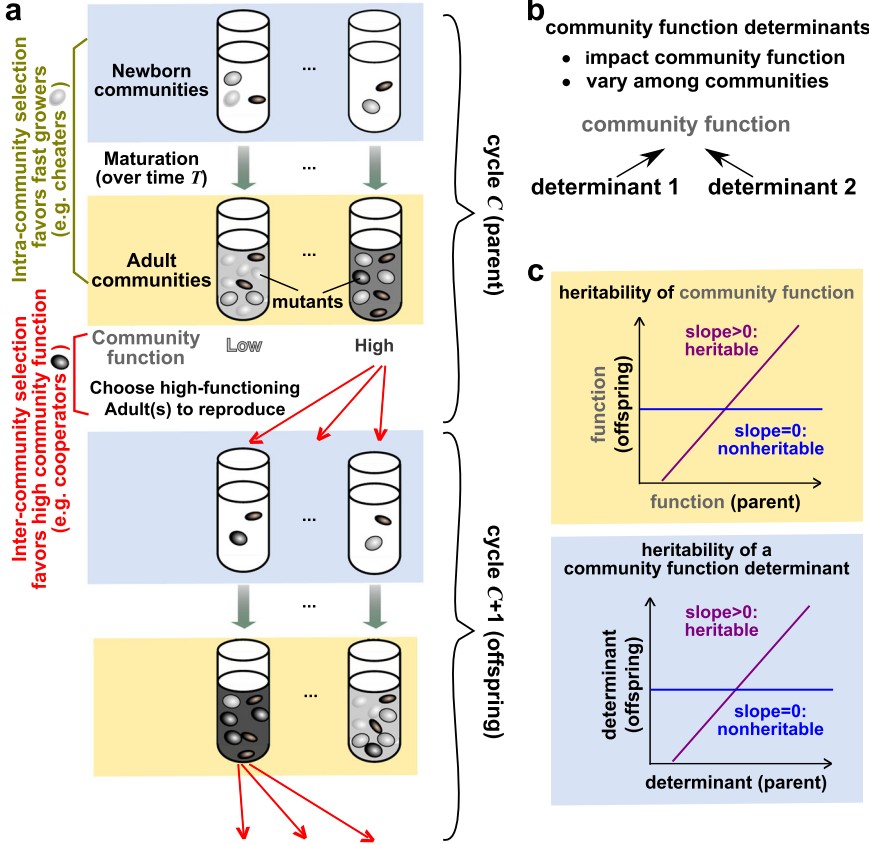

**Fig. 1 Artificial community selection to improve a community function. a** Artificial community selection. In each cycle, Newborns (blue shading) mature into Adults (yellow shading) during which intracommunity selection favors fast growers (olive bracket). The highest-functioning Adults are then chosen to reproduce, and this intercommunity selection favors high function (red bracket). Note that both parent (top half) and offspring (bottom half) communities have Newborn and Adult stages. Thus, the four rows from top to bottom represent parent Newborn, parent Adult, offspring Newborn, and offspring Adult, respectively. **b** Community function determinants. **c** Heritability of community function and community function determinants. Similar to how heritability of individual-level trait is determined[60], we estimate heritability of community function (top) and of community function determinants (bottom) from the slope of linear regression between parent functions (or determinants) and offspring functions (or determinants). Thus, although the notion of heritable versus nonheritable is convenient for conceptualization, heritability is quantitative.

and heritability of *community function determinants*. Community function determinants ("determinants") are defined as factors that determine community function and that vary among communities (Fig. 1b). Examples of community function determinants include genotype and species composition. Variation in community function is induced by variation in community function determinants. For example, mutations and species migration can introduce variation in community function by modifying genotype and species compositions. Heritability of community function[30] is determined by heritability of community function determinants. Both can be estimated from the slope of the parent–offspring linear regression (Fig. 1c). Indeed, artificial community selection could fail, unless one promotes both variation (e.g., choosing a sufficiently large number of Adult communities to reproduce) and heritability of community function (e.g., promoting species coexistence)[6,15,25,26,28].

Artificial community selection becomes particularly challenging when community function incurs a fitness cost to one or more community members. For example, fast-growing species must evolve slower growth to coexist with slow-growing partner species. If contributing to community function requires diversion of cellular resources, then within a species, contributors to community function will be outcompeted by cheaters who make little or no contributions. Hence, cheaters are favored by intra-community selection during community maturation (Fig. 1a, olive bracket). However, cheaters are disfavored by inter-community selection that only allows high-functioning communities to reproduce (Fig. 1a, red bracket). Thus, to improve a costly community function, intercommunity selection must overcome intracommunity selection (Fig. 1a).

To learn general principles on effective community selection and to gain insights that can guide future experiments, we use individual-based simulations to compare multiple selection strategies. We have conjured a highly simplified microbial community where two species coexist due to a commensal ecological interaction, and both species are required for community function. In our system, species composition is nonheritable: variations in Newborn species compositions are rapidly dampened during community maturation due to an "attractor" (steady-state species composition) induced by the commensal ecological interaction. In contrast, genotype composition is heritable. We visualize "community function landscape" relating community function to its heritable and nonheritable determinants, similar to a phenotype landscape relating an individual's phenotype to its genetic and environmental determinants[31,32]. We find that the steady-state species composition confines evolving communities to a path in the landscape. This confinement can generate counterintuitive evolutionary dynamics and prevent the attainment of maximal community function. Importantly, the local landscape geometry near the steady state species composition is indicative of community function heritability, an idea similar to phenotype landscape indicative of heritability of individual traits. When communities are trapped in low-heritability landscape regions, community function does not improve or improves only slowly under selection. Inspired by these observations, we devise perturbation strategies that improve community function heritability and thus the rate of community function improvement, even when community function landscape is high-dimensional and cannot be visualized.

## Results

### A commensal community with a species-composition attractor.
In our previous work[15], we simulated artificial selection on a two-species Helper–Manufacturer community ("H–M community", Fig. 2a). In this community, Helper (H) digests agricultural waste and consumes Resource, grows biomass, and, at no cost to itself, releases a metabolic Byproduct essential for Manufacturer (M). As M consumes Byproduct and Resource, its cellular resource is partitioned, so that a fraction $f_P$ ($0 \leq f_P \leq 1$) is used to synthesize Product P that is of interest to the experimentalist, while the rest ($1 - f_P$) is used for its own biomass accumulation. Thus, H helps M to grow, and such commensal interaction is commonly found in microbial communities[33–38].

Since M relies on H's Byproduct, H can either drive M extinct or coexist with M[15]. When M's cost $f_P$ is large, M always grows slower than H. Thus, $\phi_M$, the fraction of M biomass in a community, declines within a cycle and over cycles until M goes extinct (Fig. 2b, arrows on the right approaching the horizontal axis). When M's cost $f_P$ is moderate or small, and when we choose growth parameters of the two species properly (Table 1, Methods "Parameter choices"), H and M can coexist at a steady-state ratio (Fig. 2b, arrows on the left approaching the positive portion of the blue dashed line). Note that communities with stably coexisting strains have been engineered in the lab[39,40]. The steady-state fraction of M biomass at various cost $f_P$ forms a species-composition "attractor" (Fig. 2b, blue dashed line): at a given cost, species composition away from the attractor is pulled toward the attractor as the community matures. Because Adult composition is restricted by the attractor, compositions of offspring Newborns will also be restricted. We define this restriction on Newborn composition as "attractor-induced Newborn restrictor" ("Newborn restrictor" or "restrictor" marked by the orange line in Fig. 2d i). Note that manipulations of the Newborn restrictor will play a key role in this work.

### Visualizing community function landscape.
H–M community function is defined as the total amount of Product accumulated in an Adult community, denoted by $P(T)$ with $T$ being the community maturation time. Community function requires both species, since H supports M growth while M makes Product. Community function is not costly to H, but costly to M (cost = $f_P$). Although evaluated at the end of a maturation cycle, community function accumulates throughout the cycle and is thus sensitive to not only species genotypes (and thus phenotypes) but also initial conditions. In our models, biological parameters (i.e. cost $f_P$, growth rates, etc.) are inherited upon cell division; we thus interchangeably refer to them as genotypes or phenotypes, depending on context.

If a community is well-mixed and if the populations are clonal with deterministic dynamics (i.e., all members of a species share the same genotypes and thus phenotypes), then community function $P(T)$ can be deterministically calculated from Equations (1–7), given model parameters (species genotypes), maturation time $T$, and initial conditions. Here, we allow only M's cost $f_P$ and the fraction of M biomass in the Newborn ($\phi_M(0)$) to vary. We fix all other parameters and all other initial conditions (i.e., Newborn total biomass, excess agriculture waste that can be regarded as a constant, and initial Resource). Thus, community function has two determinants ($f_P$ and $\phi_M(0)$), and "community function landscape" can be visualized as a function of these two determinants, similar to a topographic map (Fig. 2c). The single peak corresponds to the global maximal community function achieved at an intermediate species composition and an intermediate M's cost[15]. Note that when M pays a low cost, community function is low because little Product is made, but when M pays a very high cost, community function is also low because M cannot proliferate substantially. Similarly, community function peaks at an intermediate value of Newborn species composition ($\phi_M(0)$), since community function requires both M and H.

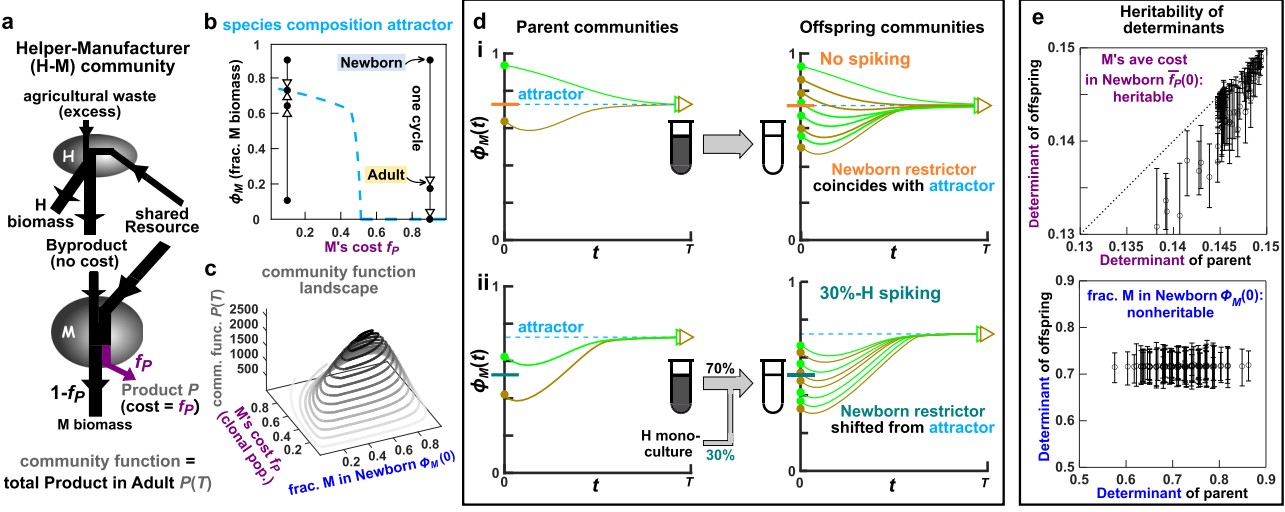

**Fig. 2 A two-species commensal community, its species-composition attractor and restrictor, and its community function landscape in relation to heritable and nonheritable determinants. a** The Helper–Manufacturer commensal community. **b** Species-composition attractor. Each arrow describes the change in $\phi_M$, the fraction of M's biomass in a community, from Newborn to Adult over one maturation cycle. All arrows are "attracted" to the species-composition attractor (blue dashed curve). Depending on whether M's cost $f_P$ is high or low, M goes extinct or coexists with H, respectively. **c** Community function landscape for clonal H and M populations. (**b**) and (**c**) are calculated by numerically integrating Equations (1–7) with Newborn's total biomass = 100, maturation time $T = 17$, and all M cells having the same cost $f_P$. Note that community function is defined for Adults, since Newborns have zero Product. Newborn (but not Adult) species composition is a community function determinant, since at a fixed cost $f_P$, different Newborn species compositions lead to the same Adult species composition (**b**) yet different community functions (**c**). **d** Newborn composition restrictor ("restrictor") coincides with the attractor, unless experimentally manipulated. (**i**) Two parent Newborns mature into parent Adults whose species compositions reach the same attractor. When these two Adults reproduce via pipetting, species compositions of offspring Newborns (green and mustard dots) fluctuate around the attractor. Therefore, the Newborn restrictor (orange) coincides with the attractor (blue). (**ii**) When 70% of Newborn biomass comes from the parent Adult and the other 30% comes from H (30%-H spiking), the Newborn restrictor (teal line) is shifted down from the attractor by about 30%. Note that the effect of spiking on species composition is erased over one cycle—Adult species compositions reach the attractor regardless of spiking. **e** Newborn average cost $\bar{f}_P(0)$ is largely heritable (positive slope), while Newborn species composition $\phi_M(0)$ is not heritable (zero slope). Individual-based stochastic simulations were carried out over one selection cycle (Methods), and M's cost $f_P$ can mutate. Each circle is obtained by plotting the mean determinant of ~60 offspring Newborns against the determinant of their parent Newborn. The corresponding error bars extend from 25% to 75% quantile (see Supplementary Fig. 4 for statistical distributions of the two determinants).

## Table 1 Parameters for genotypes (and thus phenotypes) of H and M used in the simulations.

| | Definition | Ancestral (e.g., Fig. 7) | Bounds (e.g., Figs. 2–6) |
|---|---|---|---|
| $f_P$ | fraction of M growth diverted to producing P | 0.10 | 1 |
| $K_{MR}$ | fold of $R(0)$ at which $g_{Mmax}/2$ is achieved in excess B | 1 | 1/3* |
| $K_{MB}$ | amount of Byproduct at which $g_{Mmax}/2$ is achieved in excess R | $\frac{5}{3} \times 10^2$ | $\frac{1}{3} \times 10^2$* |
| $K_{HR}$ | fold of $R(0)$ at which $g_{Hmax}/2$ is achieved | 1 | 1/5* |
| $A_0$ | in the exploitative community where M inhibits H via A (Supplementary Fig. 21), the amount of A at which H's growth rate halves | $10^3$ | $2 \times 10^3$* |
| $B_0$ | in the mutualistic community where H is inhibited by Byproduct B (Supplementary Fig. 20), the amount of B at which H's growth rate drops by $e^{-1}$ | $\frac{2}{3} \times 10^2$ | $\frac{1}{3} \times 10^3$* |
| $g_{Mmax}$ | maximal biomass growth rate of M | 0.58 per unit time | 0.7 per unit time* |
| $g_{Hmax}$ | maximal biomass growth rate of H | 0.25 per unit time | 0.3 per unit time* |
| $\delta_M$ | death rate of M | $3.5 \times 10^{-3}$ per unit time | |
| $\delta_H$ | death rate of H | $1.5 \times 10^{-3}$ per unit time | |
| $c_{RM}$ | fraction of $R(0)$ consumed per M biomass grown | $10^{-4}$ | |
| $c_{RH}$ | fraction of $R(0)$ consumed per H biomass grown | $10^{-4}$ | |
| $c_{BM}$ | amount of Byproduct consumed per M biomass grown | $\frac{1}{3}$ | |
| $P_{mut}$ | mutation probability per cell division for each mutable phenotype | $2 \times 10^{-3}$ | |

Simulations in Fig. 7 start with H and M whose genotypes (and thus phenotypes) are listed in the "Ancestral" column. These genotypes are not allowed to change beyond values listed in the "Bounds" column. In most simulations where only $f_P$ can be modified by mutations, except for $f_P$, parameters in the Bounds column are used. For maximal growth rates, and H's sensitivity to A and B ($A_0$ and $B_0$, respectively), * represents evolutionary upper bound. For $K_{SpeciesMetabolite}$, * represents evolutionary lower bound, which corresponds to evolutionary upper bound for Species's affinity for Metabolite ($1/K_{SpeciesMetabolite}$). For parameter justifications, see our previous work[15].

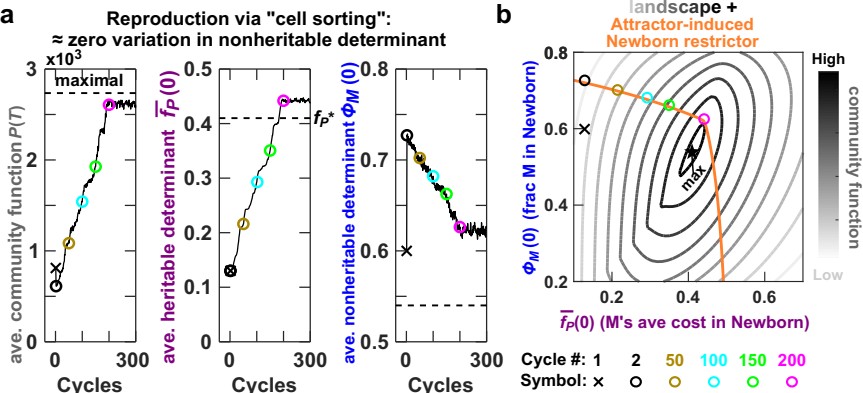

**Fig. 3 Restriction in Newborn species composition can produce counterintuitive evolutionary dynamics and prevent the attainment of maximal community function. a** Cost can evolve to be higher than what is optimal for community function. The evolutionary dynamics of the H–M community under a "cell sorting" community selection scheme (Methods) is plotted in black curves. Over each selection cycle, 100 Newborns matured into 100 Adult communities, and two Adults with the highest amount of Product were chosen to reproduce where each was randomly split into 50 Newborns. Chosen Adults were reproduced through cell sorting, and thus each Newborn had a total biomass very close to the target value of 100 and species ratio very close to that of the parent Adult. Community function determinants $\bar{f}_P(0)$ or $\phi_M(0)$ were obtained from the Newborn stage of chosen Adults, and then averaged over chosen communities. The black dashed lines mark the theoretical maximum of $P(T)$ (corresponding to the black star in **b**) and the corresponding $\bar{f}_P(0)$ and $\phi_M(0)$ required to achieve maximal community function. **b** Newborn restrictor constrains community function away from the theoretical maximum. Evolutionary dynamics of community function determinants (first cycle marked with a cross and subsequent cycles marked with circles) are plotted on top of community function landscape (gray contours) overlaid with the attractor-induced Newborn restrictor (orange curve). Compared with the landscape in Fig. 2c, the label of the x axis changes from $f_P$, the cost of the clonal M population, to $\bar{f}_P(0)$, M's average cost in a Newborn. In other words, determinants defined at the Newborn stage are predictive of the community function defined at the Adult stage. This is because we have chosen maturation time to be short, such that M's average cost does not change significantly over maturation (Supplementary Fig. 3).

The community function landscape calculated above can be used even when the populations are not clonal, as long as genotype composition does not change drastically within a cycle. Specifically, when maturation time is ~6 population doublings, community functions from individual-based simulations (where individual M cells can have different cost genotypes; Supplementary Fig. 1, i–iii; Methods) are well predicted by deterministic calculations based on Newborn species composition ($\phi_M(0)$) and the average cost paid by M in the Newborn ($\bar{f}_P(0)$) (Supplementary Fig. 3A). Overall, community function during evolution has been simplified to rely on only two determinants: Newborn species composition ($\phi_M(0)$) and Newborn genotype composition (average cost $\bar{f}_P(0)$).

**Heritability of community function determinants**. Of the two community function determinants, M's average cost in a Newborn $\bar{f}_P(0)$ is heritable, while Newborn species composition $\phi_M(0)$ is nonheritable (Fig. 2e). The heritability of genotype composition is intuitive: if a parent Newborn is dominated by cheaters (low $\bar{f}_P(0)$) or cooperators (high $\bar{f}_P(0)$), then so will its offspring Newborns (Fig. 2e top). Note that offspring costs are generally less than the parent cost, since cooperator frequency declines during community maturation due to cheater takeover (circles are below the dotted line of slope 1 in Fig. 2e top). Newborn species composition $\phi_M(0)$ is not heritable due to the attractor. Parent Newborns experience stochastic fluctuations in species composition (e.g., a species ratio of 50:50 can become 40:60 by chance due to pipetting a small number of cells). However, due to the attractor, parent Adults end up sharing similar species compositions (Fig. 2d i, left), and so will their offspring Newborns (Fig. 2d i, right). In essence, variations in Newborn species composition are not transmitted across cycles, and are therefore not heritable (Fig. 2e bottom). Since any elevation in community function due to nonheritable determinant will not transmit to the next cycle, we can quantify selection efficacy as the progress in the heritable determinant $\bar{f}_P(0)$.

**Restrictor leads to counterintuitive and suboptimal outcomes**. We now consider the case where ancestral M pays a cost smaller than what is optimal for maximal community function. This scenario poses a common problem that is challenging to address: while maximal community function requires a higher cost, intracommunity selection favors a lower or no cost (Fig. 1a).

To obtain selection dynamics, we performed individual-based stochastic simulations (Methods, Supplementary Fig. 1). In each cycle, we select from 100 communities. To discourage cheater takeover, each Newborn has a small total biomass (100 biomass units or 50 ~ 100 cells; Supplementary Fig. 2), and Newborns mature into Adults over a relatively short period of time ($T = 6 \sim 7$ doublings[15]). We track individual H and M cells as they consume and release metabolites, grow biomass and divide, and stochastically die. As an M cell divides, the $f_P$ of both daughters have a probability (0.002/cell/generation) to mutate, with 50% of mutations setting $f_P$ to 0, while the rest increasing or decreasing $f_P$ by on average 5–6 percent. M cells with a higher $f_P$ contribute more toward community function but grow slower. At the end of $T$, Adults are ranked on their functions. The top 2 or 10 Adults are chosen to reproduce, with their H and M cells randomly distributed into offspring Newborns so that Newborn total biomass fluctuates around the target value (100 biomass units or 50–100 cells) and Newborn species compositions fluctuate around that of the parent Adult. We do not mix different community lineages to preserve variations among communities and to prevent cheaters from spreading across communities (Supplementary Fig. 5). Our choices of model parameters are supported by the microbial experimental literature (Methods, Parameter choices).

In the absence of community selection (e.g., allowing each Adult to reproduce one offspring, or randomly selecting Adults to reproduce), then unsurprisingly, community function rapidly declines to zero as cheaters take over[15].

To ensure effective community selection, we increase community function heritability by reducing variations in the

nonheritable determinant $\phi_M(0)$. To do so, we reproduce Adults via "cell sorting", as if by a flow sorter capable of measuring the biomass of individual H and M cells, so that all Newborns from the same parent have nearly identical species composition and total biomass. With this experimentally challenging method, community selection successfully improves community function over ~200 selection cycles (Fig. 3a). However, community function $P(T)$ never reaches the theoretical maximum (Fig. 3a dashed line). The average cost paid by M stays above $f_P^*$ optimal for community function (Fig. 3a, middle panel), which is surprising because high cost would be disfavored by both intracommunity selection (which favors fast growth and low cost) and intercommunity selection (which favors $f_P^*$).

To understand this suboptimal selection outcome, we overlay community function landscape (Fig. 3b gray contours) with attractor-induced Newborn restrictor (Fig. 3b orange line, which coincides with the attractor—see Fig. 2d i). We note that the Newborn restrictor does not pass through the maximal community function (black star in Fig. 3b). During selection, Newborn species compositions are strictly confined to the restrictor (Fig. 3b, circles on top of the orange line). This explains the suboptimal selection outcome: Like a hiker who is restricted to a trail that does not traverse the mountain top, community function can only climb to the highest value along the restrictor (magenta circle), which is lower than the global maximum (black star). Consequently, the corresponding Newborn average cost $\overline{f}_P(0)$ and Newborn species composition $\phi_M(0)$ are higher than those optimal for community function. Not surprisingly, community function can reach the maximum if we push down the Newborn restrictor by an appropriate amount (e.g., by replacing 15% of Newborn total biomass with H during each cycle of selection), as shown in Supplementary Fig. 8. We will not dwell on maximal community function further, since the maximal function is typically unknown in practice, as in many global optimization problems. Instead, we will now focus on the rate of community function improvement.

**Landscape determines heritability and selection efficacy.** Successful selection requires heritability in community function. Community function heritability depends on (1) variations in the heritable and nonheritable determinants, and (2) how strongly variations in these determinants impact variations in community function. Minimizing variations in the nonheritable determinant $\phi_M(0)$ (e.g., reproducing the chosen Adults through cell sorting, Fig. 3) results in high heritability in community function and thus rapid improvement under selection[15]. However, such a technique is often difficult to apply.

We now explore how to achieve rapid improvement in community function when variations in nonheritable determinants cannot be easily diminished. We will show that the local geometry of community function landscape predicts the heritability of community function and thus the efficacy of intercommunity selection. To illustrate, we consider a cartoon model in Fig. 4 where community contours are straight lines. In Fig. 4a, community function contours are perpendicular to the axis of the heritable determinant. Therefore, variation in community function (light to dark gray) is fully attributed to variation in the heritable determinant. Thus, community function is heritable (Fig. 4d), and intercommunity selection can make large progress in the heritable determinant over a selection cycle (Fig. 4a). In contrast, when community function contours are parallel to the axis of the heritable determinant (Fig. 4b), no variation in community function can be attributed to variation in the heritable determinant. Thus, community function is

nonheritable (Fig. 4e), and intercommunity selection makes no progress (Fig. 4b). An intermediate case is shown in Fig. 4c and f.

**Boosting heritability hastens community function improvement.** We can now examine community selection when the nonheritable determinant (Newborn's fraction of M biomass $\phi_M(0)$) is allowed to fluctuate during community reproduction. Specifically, if we simulate pipetting cells from Adults to seed Newborns (while keeping the total Newborn biomass fixed), $\phi_M(0)$ will fluctuate stochastically (due to small Newborn size), essentially creating a "cloud" of Newborns around the Newborn restrictor (Fig. 5b ii). Each Newborn's community function at Adulthood can be read out from the value (gray shade) of the contour it resides on. Although landscape contours near the restrictor are not straight lines, they are largely parallel to the heritable determinant axis (similar to Fig. 4b). Thus, variations in community function are largely attributed to variations in the nonheritable determinant. Indeed, community function has low heritability (~0 slope in Fig. 5c ii), and intercommunity selection makes only a small progress in the heritable determinant (short red arrow in Fig. 5b ii; statistics in the red box plot of Fig. 5b iv). This small progress is just enough to counter the decline due to intracommunity selection (olive box in Fig. 5b iv), resulting in a net of zero-improvement rate (black box in Fig. 5b iv). Since heritability remains low from cycle to cycle (Supplementary Fig. 23 top panels), community function and heritable determinant barely improve despite 1000 selection cycles (Fig. 6a).

We then seek to improve selection by shifting the Newborn restrictor to a location with higher heritability, i.e., where community function contours are largely perpendicular to the axis of heritable determinant (e.g., south of the orange restrictor). This can be achieved by replacing 30% of total Newborn biomass with nonevolving H ("30%-H spiking", teal box in Fig. 5a i, Fig. 2d ii). Under 30%-H spiking, community function becomes more heritable (positive slope in Fig. 5c iii) due to reduced dependency on the nonheritable determinant and enhanced dependency on the heritable determinant (Supplementary Fig. 6). Intercommunity selection thus makes a larger improvement in the heritable determinant (compare red arrows in Fig. 5b iii versus ii; compare red boxes in Fig. 5b v versus iv), and the total improvement rate becomes positive (black box in Fig. 5b v). Since heritability fluctuates around a high level from cycle to cycle (Supplementary Fig. 23 bottom panels), community selection improves community function efficiently (Fig. 6b). Spiking must be performed at each cycle since spiked species composition rapidly returns to the attractor (Fig. 2d ii).

Under a variety of conditions, community function improves faster when community function heritability is enhanced through species spiking. The 30%-H spiking strategy boosts the already increased selection efficacy when we choose top-10, instead of top-2 Adults, to reproduce (Fig. 6: c more effective than a due to increased variation among communities[15]; d more effective than c due to spiking and consequently improved heritability). H spiking promotes selection when measurement noise in community function interferes with selection (Fig. 6: f better than e, h better than g), and when both total biomass and species composition of Newborns are allowed to fluctuate as if the chosen Adults are reproduced and spiked through pipetting without fixing the inoculum biomass (Supplementary Fig. 10). Although quantitative details differ, H spiking at a wide range of percentages speeds up the improvement of community function (Supplementary Fig. 10). Importantly, compared with the ancestral community, evolved communities selected under one maturation time ($T = 17$) exhibit higher functions across a range

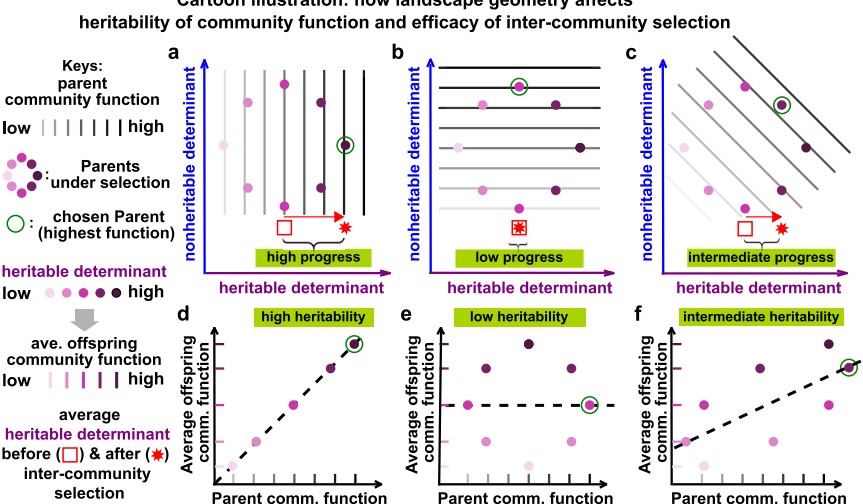

**Fig. 4 The local geometry of community function landscape dictates the heritability of community function and consequently the efficacy of intercommunity selection.** Eight parent communities (dots) are under selection. Parent community function can be read from the gray shade of the underlying contour where a darker shade indicates a higher function (top panels). Parent's heritable determinant (purple shade of the dot) dictates the average function of its offspring communities (as effects from the nonheritable determinant average out). Thus, we can plot parent function (gray shade) against average offspring function (purple shade) (bottom panels), and the slope of the least-square linear regression (black dashed line) is an estimation of community function heritability. The community with the highest function (sitting on the darkest contour) is chosen for reproduction (green open circle). Average heritable determinants before and after intercommunity selection are marked by a red square and a red star, respectively. Thus, progress made by intercommunity selection can be measured as the distance from red square to red star. Intercommunity selection is the most or the least effective when community function contours are perpendicular (**a**, **d**) or parallel (**b**, **e**) to the axis of heritable determinant, respectively, and intermediate otherwise (**c**, **f**).

of maturation times (from $T = 13$ to 21) and Newborn species compositions (Supplementary Fig. 7).

Species spiking is but one of the perturbations we can use to alter community function heritability. For example, if we extend maturation time $T$ from 17 to 20 (and assume that any potential resource depletion will not affect cell phenotypes), then community function heritability is improved (larger slope in Supplementary Fig. 9c than in Fig. 5c ii). This leads to a faster improvement of community function (compare Supplementary Fig. 9a with Fig. 6a).

**Enhancing selection efficacy without knowing the landscape.** So far, we have examined a simple scenario where community function landscape can be visualized. Since landscapes of most community functions are high-dimensional and unknown, it is infeasible to devise a perturbation strategy based on landscape visualization. However, landscape geometry is reflected in the heritability of community function (Fig. 4; Fig. 5b and c), which can be estimated from experimental measurements (similar to Fig. 5c). Thus, we can try several different perturbation strategies, compare them, and choose the strategy yielding the highest community function heritability. Since communities move on the landscape as they evolve, periodic heritability check is needed.

As an example, let us consider a more complex scenario with the H–M community. If we allow growth parameters of H and M to also evolve, then community function will have six heritable determinants all defined at the Newborn stage (Supplementary Fig. 3b): M's average cost, the average maximal growth rates of H and of M, the average affinities of M to Resource and to Byproduct, and the average affinity of H to Resource. If we reproduce the Adult via "pipetting", then Newborn's total biomass and species composition fluctuate stochastically, adding two nonheritable determinants. If we additionally consider community function measurement noise (a normal random variable with mean 0 and standard deviation comparable to the ancestral

community function), we have yet another nonheritable determinant. Overall, community function now has nine determinants, six heritable and three nonheritable.

We simulate community selection in the above complex scenario (schematic in Supplementary Fig. 11). We start with the no-spiking strategy, and always choose top-10 Adults where each reproduces 10 Newborns. If a fraction of Newborn biomass is to be replaced with H (or M) biomass, the spiking mix consists of equal parts of five evolved H (or M) clones randomly isolated from the previous cycle of the same lineage (Supplementary Fig. 11a). During reproduction, portions of a chosen Adult and the spiking mix are "pipetted" to initiate Newborns, so that both the total biomass and the species composition fluctuate stochastically in Newborns. Every 100 cycles, we update the spiking strategy based on heritability of community function. Specifically, we quantify community function heritability for five candidate spiking strategies (no spiking, 30%-H spiking, 60% H spiking, 30% M spiking, and 60% M spiking) by regressing parent function with median offspring function (similar to Fig. 5c). The current spiking strategy is then updated if an alternative strategy confers significantly higher community function heritability (Supplementary Fig. 11b, Methods).

With the no-spiking strategy, community selection moderately improves community function and heritable determinants (Fig. 7a and b; Supplementary Fig. 12a). The rate of community function improvement is higher when the spiking strategy is periodically adjusted according to community function heritability (Fig. 7c and d; Supplementary Fig. 12b). In contrast, adopting the spiking strategy with the lowest heritability leads to a slower rate of improvement (Supplementary Fig. 12c), while randomly choosing spiking strategy leads to variable results (Supplementary Fig. 12d). It is also noteworthy that under periodic heritability check, the adopted spiking strategy is not static (Fig. 7e). A static 60% H or 30%-H spiking strategy offers negligible improvement over no spiking (Supplementary Fig. 13). Communities obtained through selection with periodic

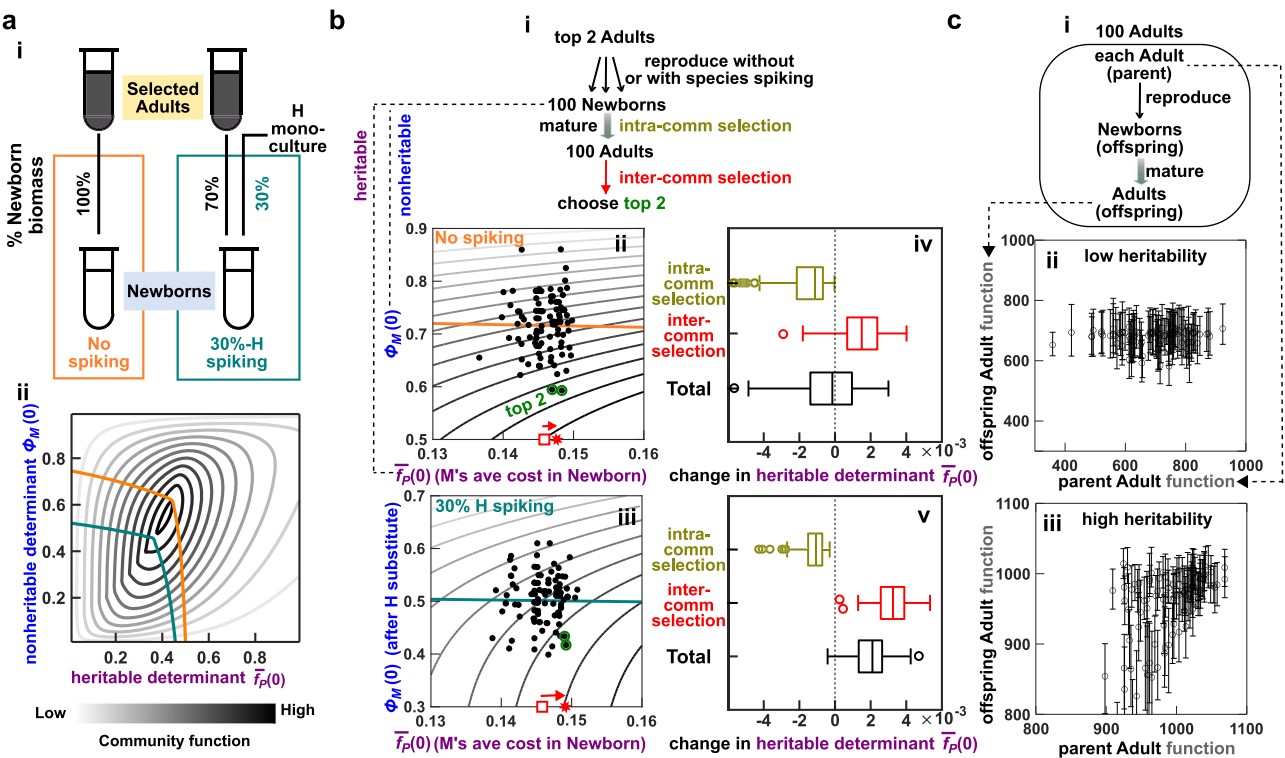

**Fig. 5 Increasing community function heritability improves selection efficacy. a** Newborn restrictor (orange curve in ii) is shifted down when 30% of the total biomass in a Newborn is replaced by the nonevolving H ("30%-H spiking"; teal curve in ii) in every cycle. Because communities are constrained by the Newborn restrictor, we only need to focus on the geometry of the landscape near the restrictor. **b** Improving community function heritability improves selection efficacy. **i**: Scheme of community selection over one cycle. **ii, iii**: Visualization of intercommunity selection on the landscape under the no-spiking strategy and the 30%-H spiking strategy, respectively. The 30%-H spiking strategy is expected to improve community function heritability, since contours near the teal line are more perpendicular to the axis of heritable determinant than contours near the orange line. The top 2 communities chosen to reproduce are highlighted with green circles. The average heritable determinant $\bar{f}_P(0)$ over all 100 communities and that over the top 2 are indicated by red open square and red star, respectively. The length of the red arrow indicates the progress in heritable determinant $\bar{f}_P(0)$ due to intercommunity selection. **iv, v**: Progress in the heritable determinant $\bar{f}_P(0)$. In total, 100 repeats of the selection cycle are used to obtain box plots (center: median; bottom and top edges: the 25th and 75th percentiles, respectively; whiskers: data range excluding outliers; open circles: outliers). Although intercommunity selection (red) improves heritable determinant $\bar{f}_P(0)$, intracommunity selection (olive) reduces it. The sum of these two effects is the net progress (black). **c** Species spiking can improve the heritability of community function. **i**: Heritability is quantified by comparing 100 pairs of parent Adult functions and their offspring Adult functions and estimating the slope of the least-squares linear regression. **ii**: Under the no-spiking strategy, community function heritability is low. **iii**: Under the 30%-H spiking strategy, community function heritability is high. Circles represent the mean, and error bars extend from 25% to 75% quantile. In (ii) and (iii), error bars are calculated from ~60 to 100 offspring communities, respectively. Empirical determination of heritability as depicted here can be used in lieu of landscape visualization to guide perturbation strategy.

heritability check exhibit higher functions than those obtained through no spiking, even if Newborn species compositions are readjusted to values over a wide range (Fig. 7f). Therefore, it is important to evaluate heritability periodically and update perturbation strategy accordingly.

**Increasing heritability as an effective and general approach.**
Qualitatively similar results are obtained when the number of colonies used to make the spiking mix is changed from 5 to 1, 2, or 10 clones (Supplementary Figs. 14, 15 and 16). Simulations with three candidate spiking strategies (no spiking, 30% M spiking, and 30%-H spiking) instead of five strategies generate qualitatively similar results (Supplementary Fig. 22). The frequency of heritability check can be adjusted. For example, similar outcomes are obtained if heritability checks are performed "adaptively" (e.g., only when the average rate of community function improvement over the last 50 cycles is less than zero, Supplementary Fig. 17). In this particular case, adaptive check reduces the number of checks by ~50% compared with Fig. 7 (periodic check every 100 cycles).

Heritability checks can also speed up community function improvement for communities engaging in mutualistic and exploitative ecological interactions, even when the community function landscape is high-dimensional and unknown. First, we simulated community selection on a mutualistic H–M community, where M relies on H's Byproduct, and H's Byproduct inhibits H's growth. By removing Byproduct, M promotes H's growth. Similar to the simulations shown in Fig. 7, genotypes that can be modified by mutations include M and H's maximal growth rates, affinities to metabolites, and M's cost $f_P$. Additionally, H's sensitivity to its Byproduct can also be modified by mutations. At the end of each cycle, Adult communities with top-10 functions are chosen and each reproduces 10 offspring Newborn communities through pipetting for the next cycle. The community function of this mutualistic H–M community thus has seven heritable determinants and three nonheritable determinants. Improvement in community function is rapid under selection even without spiking, and is sped up slightly but significantly (Mann–Whitney U test, $n_1 = n_2 = 6$, $p = 10^{-3}$, one-tailed) when we periodically adopt the spiking strategy conferring the highest community function heritability (Supplementary Fig. 20). Next,

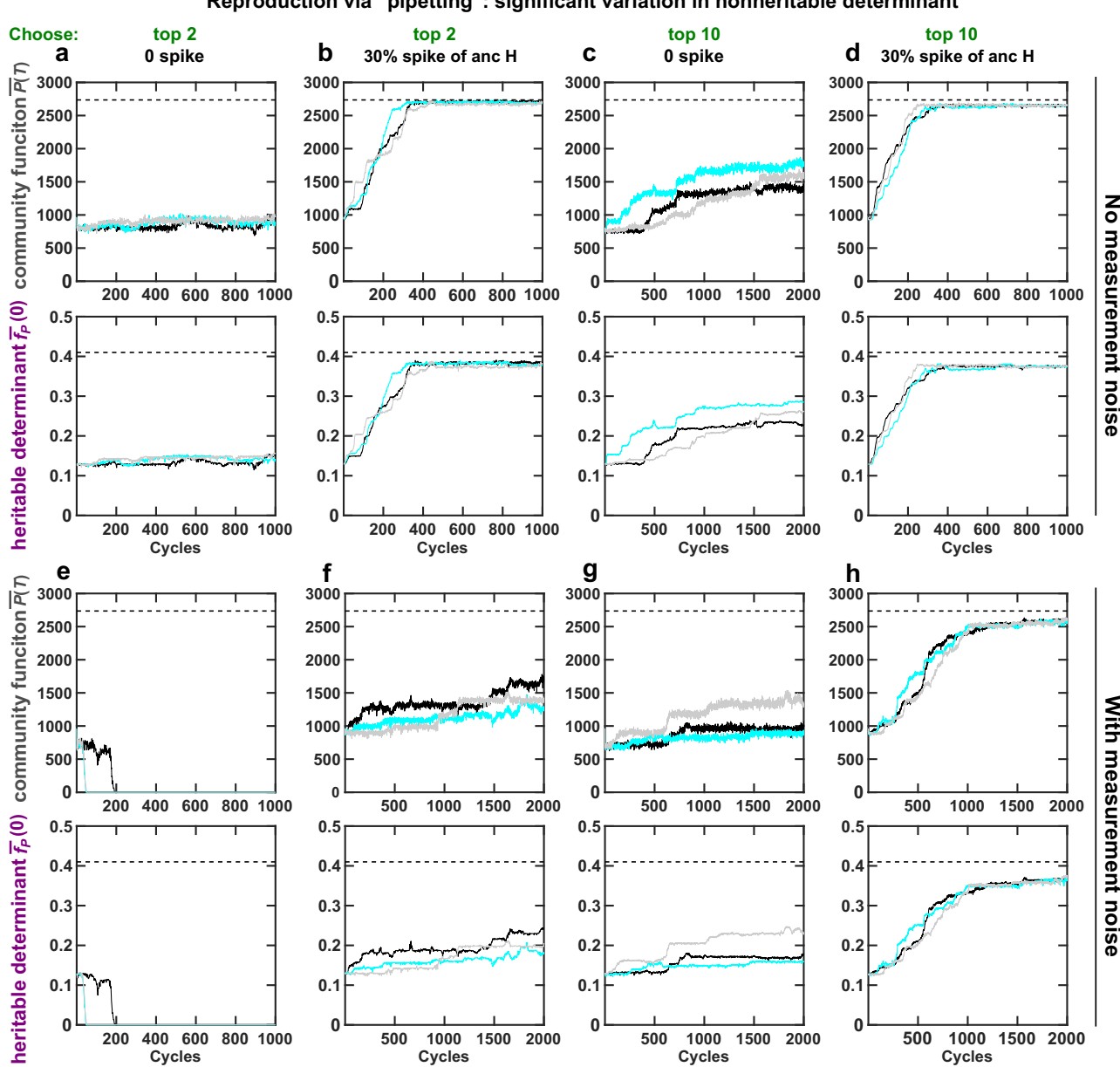

**Fig. 6 Increasing community function heritability by shifting Newborn restrictor can improve selection efficacy under various conditions.** Evolutionary dynamics of community function $P(T)$ and its heritable determinant $\overline{f}_P(0)$ averaged over the chosen communities under different selection schemes are plotted. Species composition is allowed to stochastically fluctuate during community reproduction, as if pipetting an inoculum of a fixed total biomass from an Adult to seed its Newborns. In all cases, community selection is more effective under the 30%-H spiking strategy (**b**, **d**, **f**, and **h**, where 30% of Newborn biomass is replaced by the nonevolving H) compared with the no-spiking strategy (**a**, **c**, **e** and **g**). In "top-2" strategy (**a**, **b**, **e** and **f**), top-2 communities each reproduces 50 Newborns. In "top-10" strategy (**c**, **d**, **g** and **h**), top 10 communities each reproduces 10 Newborns. In "no measurement noise" (**a**–**d**), the measured community function is the true community function obtained in the simulation. In "with measurement noise" (**e**–**h**), measured community function is the sum of the true community function and a normal random variable with a mean of 0 and a standard deviation of 100. Dashed lines correspond to theoretical maximal community function or M's cost $f_P^*$ optimal for community function. Black, cyan, and gray curves represent three independent replicates.

we simulated community selection on an exploitative H–M community. In this community, M releases a compound that inhibits H. That is, H helps M, but M inhibits H. Since H's sensitivity to the compound released by M can be modified by mutations, this exploitative community also has seven heritable determinants and three nonheritable determinants. Compared with selection with no spiking, the rate of community function improvement is much faster when we periodically adopt the

spiking strategy conferring the highest community function heritability (Supplementary Fig. 21).

**Discussion**

We start with a highly simplified case where the community function of interest varies due to variations in two determinants—one heritable (genotype composition) and one nonheritable (species composition that can change rapidly due to ecological

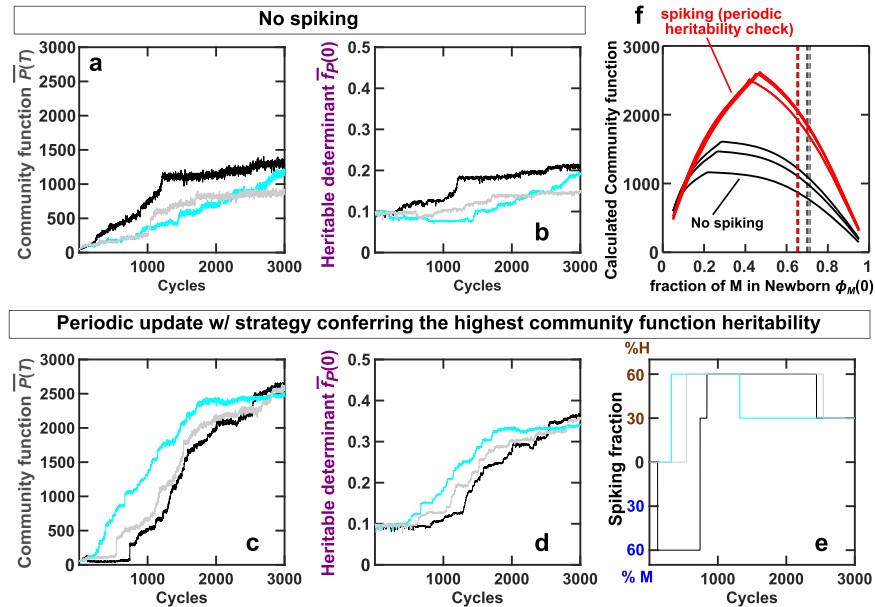

**Fig. 7 Adjusting perturbation strategies based on periodic heritability check improves selection efficacy even when the landscape is high-dimensional and unknown.** Plots show the evolutionary dynamics of community function $P(T)$ and the heritable determinant $\bar{f}_P(0)$ averaged over chosen communities under the no-spiking strategy (**a**, **b**) and under spiking strategy dictated by periodic heritability check (**c**, **d**). During the periodic check, community function heritability under five candidate spiking strategies (no spiking, 30%-H spiking, 60% H spiking, 30% M spiking, and 60% M spiking) is evaluated every 100 cycles. The current strategy is replaced by an alternative strategy if the alternative strategy confers significantly higher heritability. Spiking mix consists of five evolved H clones or five evolved M clones randomly isolated from the same community lineage in the previous cycle. The corresponding spiking percentages determined by periodic check are displayed in (**e**). Black, cyan, and gray curves represent three independent simulation replicas. Due to the high-dimensional landscape, we can no longer determine the theoretical maximal community function. **f** Communities evolved under periodic heritability check (red) exhibit higher functions than communities evolved under no spiking (black). To compare communities evolved under these two strategies, we analyze communities with the highest functions in Cycle 3000 of simulations in (**a**) and (**c**), and plot their functions calculated using Equations (1–7) at various Newborn species $\phi_M(0)$, based on their heritable determinants ($\bar{f}_P(0)$, $\bar{g}_{Mmax}(0)$, $\bar{K}_{MR}(0)$, etc.). We also calculated the steady state $\phi_M$ from the same equations and plot them in vertical dashed lines.

interactions). These two determinants capture crucial elements common to community function in general. Hence, our work serves as a fundamental building block for conceptualizing community selection, akin to physicists studying the ideal gas, or population geneticists studying a single-locus two-allele trait. Below, we recap our work's major conclusions, assumptions, limitations, and generality. Then, we reflect on what makes community selection effective, and discuss future directions.

We have demonstrated how ecological interactions, especially those that encourage species coexistence, can result in a species-composition attractor that impacts community selection. The attractor restricts Newborn species compositions to a narrow region, which can (1) generate counterintuitive evolutionary dynamics (Fig. 3a middle panel); (2) constrain communities away from maximal community function (Fig. 3b); and (3) trap communities in a region of low heritability where selection efficacy is low (i.e., community function improves slowly or not at all despite selection; Figs. 5c ii and 6a). This understanding has helped us to devise perturbation strategies that improve community function heritability and in turn selection efficacy (Figs. 5c iii and 6b, Supplementary Fig. 9). When we can visualize the landscape, we can identify high-heritability regions where community function contours are perpendicular to the axis of heritable determinant (Fig. 4). We can then design perturbation strategies (e.g., species spiking and varying maturation time) to shift the restrictor to these regions (Fig. 5). When we can not visualize the landscape, we can choose perturbation strategies based on community function heritability (Fig. 7), a quantity that can be estimated from experiments (similar to Fig. 5c). Since

evolving communities move in the landscape, perturbation strategy needs to be adjusted—either periodically (Fig. 7; Supplementary Fig. 11) or whenever selection progress slows down (Supplementary Fig. 17).

Our study assumes no spatial structure within communities, the presence of a species-composition attractor, and time scale separation (ecological dynamics much faster than evolutionary dynamics within a cycle). These assumptions capture realistic situations: spatial structure can be disturbed by convection-induced mixing, and by fast diffusion of metabolites if communities are encapsulated in small droplets. Composition attractor and thus Newborn restrictor can arise from ecological interactions that benefit at least one species, or when different species show distinct environmental preferences[16,35,37,41–46]. Ecological timescale is faster than evolutionary timescale because mutation rate is small (our mutation rate of 0.002/cell/generation is already on the high end among the published values) and because community maturation time needs to be short to prevent cheater takeover[15]. Quantification of heritability does require variation in community function, or else parent–offspring regression can become a single dot and heritability becomes ill-defined. One such example is found in the theoretical work by Doulcier et al.[16] where species ratio evolves to a target value and is thus identical among all evolved communities and their offspring communities. However, most published experimental work does show large variations in community functions[17–28], at least initially.

Our work has limitations and caveats. First, for species spiking, member species need to be culturable, so that we can isolate clones to form spiking mixes. Flow sorting could potentially

bypass this problem if member species can be distinguished by light-scattering or fluorescence patterns. Second, evaluating community function heritability is resource-intensive. Thus, checking heritability only when necessary (Supplementary Fig. 17) may help reduce workload. Third, the effect of perturbation may be erased every cycle (Fig. 2d ii). However, improved selection efficacy yields desired genotypes that can be frozen and repeatedly revived and used. These evolved genotypes lead to higher community functions even when maturation time or Newborn species composition differs from those during the evolution experiment (Fig. 7f and Supplementary Fig. 7). Fourth, extreme perturbations (e.g., extreme spiking percentages) can induce spurious heritability. For example, when we consider only three candidate strategies (no spiking, 30%-H spiking, and 30% M spiking), selection efficacy can be higher than when we also include the 60% H spiking and 60% M spiking strategies. In Supplementary Fig. 22a, the last stretch of community function ascent closely follows the switching of spiking strategy from 60% H to 30% H. This is presumably because 60% H spiking was so extreme that species composition does not return to the attractor within one cycle (Supplementary Fig. 24). Consequently, stochastic fluctuations in species composition become partially heritable (similar to Supplementary Fig. 19c), misleading the choice of spiking strategy.

We have tested the generality of our work in several ways. We varied the nature of perturbation (species spiking in Figs. 5–7, altering maturation time in Supplementary Fig. 9), the number of clones used in the spiking mixture (Supplementary Figs. 14, 15 and 16), the number of spiking strategies under heritability check (Supplementary Fig. 22), the frequency of heritability check (Supplementary Fig. 17), and ecological interactions between species (Supplementary Figs. 20 and 21). In all cases, improvements in community function can be sped up by perturbations that improve community function heritability. Although we mainly studied a costly community function, we obtained similar results when community function is not costly (Supplementary Fig. 18). Selection can be sped up by improving community function heritability even when species composition fails to reach the steady state by the end of maturation, in which case an otherwise nonheritable determinant becomes partially heritable (Supplementary Fig. 19). In sum, for communities with a stable species composition, appropriate perturbation strategies under the guidance of heritability checks can speed up the improvement of community function.

Our most simplified case (Figs. 5 and 6) involves a single-peaked landscape (Fig. 2c). Even if the landscape has multiple peaks, we only need to consider the landscape region near the Newborn restrictor. Should the restrictor traverse multiple peaks, we have a multipeak optimization problem. The broader optimization field deals with this by iteratively invoking an algorithm designed for a single peak, but at different starting points (e.g., MATLAB's GlobalSearch and MultiStart algorithms). In the case of community selection, this means sampling diverse starting compositions.

What makes community selection effective? Effective community selection relies on optimizing intercommunity variation, selection strength, and community function heritability. Experiments showed that low heritability of community function could limit selection efficacy[26–28]. Since our community function is affected by Newborn species composition, having an attractor does not solve the problem of nonheritable variations in Newborn species composition (contrary to the claim by Doulcier et al.[16]). Our work identifies two strategies for improving community function heritability. One strategy is to reduce the variation in nonheritable determinant $\phi_M(0)$, for example, through cell sorting[15]. Although nonheritable variation can also be reduced by

increasing Newborn size, large Newborn size leads to cheater takeover in all communities and consequently selection failure (Supplementary Fig. 2)[15]. Intriguingly, large Newborn size hinders community selection even for noncostly community function, presumably because large Newborn size reduces intercommunity variation (Supplementary Fig. 18b). The other strategy for improving heritability is to reduce the dependence of community function on the nonheritable determinant. This can be achieved through proper perturbations (e.g., manipulating species composition in Figs. 5–7 and Supplementary Fig. 6; altering maturation time in Supplementary Fig. 9). Overall, selection efficacy is improved when we improve community function heritability (compare improvement rates in Fig. 6a with Fig. 3a and Fig. 6b, Figs. 5–7). In contrast, if we reduce community function heritability (and variation) by mixing Adults before reproduction, selection efficacy is poor (Supplementary Fig. 5).

Optimizing community selection is challenging, partly because variation, selection strength, and heritability are interconnected. For example, strong selection can diminish selection efficacy by reducing intercommunity variation ("top 2" working less well than "top 10" in Fig. 6[15]). Drastically diluting a parent Adult (strong bottleneck) increases intercommunity variation, but also creates large stochastic variations that reduce heritability. Our current work showcases an additional difficulty in achieving effective community selection. On the one hand, species-composition attractor can enhance heritability of community function by promoting species coexistence. On the other hand, the attractor and its associated Newborn restrictor mean that communities may only sample a small region in the community function landscape. This can constrain both selection dynamics (Fig. 5) and selection outcome (Fig. 3b). These concepts have prompted us to devise perturbation strategies to shift the Newborn restrictor to a region conferring higher heritability.

For future directions, we start from the empirical side. (1) How to balance variation with heritability during community selection? In Chang et al.[4], selection efficacy was improved by alternating community perturbations (whose stochastic effects boost intercommunity variation but reduce heritability) with community stabilization (so that stabilized communities might display attractors). Our work suggests that following stabilization, applying perturbation strategies to increase heritability could further increase selection efficacy. (2) How to reduce workload of community selection? (3) What general principles might we learn from applying selection to diverse types of communities? (4) How to scale up evolved communities for industrial applications? Unlike community selection, large-scale productions involve large microbial populations and long duration, and these conditions will facilitate cheater takeover[15] (Supplementary Fig. 2). To combat cheaters during large-scale production, several strategies can be deployed. If the ability to make a product is engineered, then the costly synthesis of the product can be induced at the end of growth phase, thus reducing the growth advantage of cheaters. Additionally, biosensors can be engineered to link cellular growth to product level, thus blocking cheater growth[47]. Moreover, community selection can be sped up by using strains with high mutation rates (i.e., mutators), and after the desired genotype has been obtained, the mutator genotype can be repaired, so that fewer cheaters arise during large-scale production. Finally, a spatially structured environment (such as microwells or droplets) can be introduced to reduce Newborn size and restrict cheater takeover[48,49].

We also need new theories on community selection. Over the last century, a rich body of theory has been developed to understand evolution of quantitative traits in individual organisms (e.g., the work by Lynch and Walsh[50]). Two of the key concepts we use here

are landscape and heritability, which are fundamental for understanding the evolution of individual traits. For example, phenotype landscape as a function of genetic and environmental determinants was used to illustrate the evolution of developmental interactions[31,32]. The local geometry (gradient) of landscape determines how sensitive the phenotype is to underlying variations, and thus to selection based on the phenotype. Heritability is a pivotal concept in evolutionary biology, particularly in breeding. Multiple statistical methods have been developed to estimate heritability and facilitate designing effective breeding schemes[50]. These methods, although providing inspirations for this work, will need to be expanded to be directly applicable to community selection. Thus, it will be important to develop new theories that incorporate unique features of community selection, such as ecological dynamics resulting from species interactions, interactions between ecological and evolutionary dynamics, and the interplay between intra- and intercommunity selection.

## Methods

**Calculating landscape, attractor, and restrictor.** In this work, we considered communities with commensal, mutualistic, and exploitative interactions. Below, we describe the differential equations for each type of interaction, and how we calculate the corresponding community function landscape, species-composition attractor, and Newborn restrictor.

Commensal H–M community: The model community for most simulations is the same commensal H–M community used in our previous work[15]. The community function landscape plots $P(T)$ as a function of $\phi_M(0)$ and $\overline{f}_P(0)$. Assume that a Newborn community has 100 biomass units, that all cells have the same genotype (all M cells have the same $f_P = \overline{f}_P(0)$), that death and birth processes are deterministic, and that there is no mutation. $P(T)$ can then be numerically integrated from the following set of scaled differential equations for any given pair of $\phi_M(0)$ and $\overline{f}_P(0)$[15]:

$$\frac{dR}{dt} = -c_{RM}g_M M - c_{RH}g_H H \tag{1}$$

$$\frac{dB}{dt} = g_H H - c_{BM}g_M M \tag{2}$$

$$\frac{dP}{dt} = f_P g_M M \tag{3}$$

$$\frac{dH}{dt} = g_H H - \delta_H H \tag{4}$$

$$\frac{dM}{dt} = g_M(1 - f_P)M - \delta_M M \tag{5}$$

where

$$g_H(R) = g_{Hmax}\frac{R}{R + K_{HR}} \tag{6}$$

$$g_M(R, B) = g_{Mmax}\frac{R_M B_M}{R_M + B_M}\left(\frac{1}{R_M + 1} + \frac{1}{B_M + 1}\right) \tag{7}$$

and $R_M = R/K_{MR}$ and $B_M = B/K_{MB}$. Unless otherwise specified, landscapes in this paper are obtained by integrating Equations (1–5) from $t = 0$ to $t = 17$.

Equation (1) states that Resource $R$ is depleted by biomass growth of M and H, where $c_{RM}$ and $c_{RH}$ represent the amount of $R$ consumed per unit of M and H biomass, respectively. Equation (2) states that Byproduct $B$ is released as H grows, and is decreased by biomass growth of M due to consumption ($c_{BM}$ amount of B per unit of M biomass). Equation (3) states that Product P is produced as $f_P$ fraction of potential M growth. Equation (4) states that H biomass increases at a rate dependent on Resource $R$ in a Monod fashion (Equation (6)) and decreases at the death rate $\delta_H$. Note that Agricultural waste is not a state variable here as it is present in excess. Equation (5) states that M biomass increases at a rate dependent on Resource $R$ and Byproduct $B$ according to the Mankad and Bungay model (Equation (7))[51] discounted by $(1 - f_P)$ due to the fitness cost of making Product, and decreases at the death rate $\delta_M$. In the Monod growth model (Equation (6)), $g_{Hmax}$ is the maximal growth rate of H and $K_{HR}$ is the $R$ at which $g_{Hmax}/2$ is achieved. In the Mankad and Bungay model (Equation (7)), $K_{MR}$ is the $R$ at which $g_{Mmax}/2$ is achieved when $B$ is in excess; $K_{MB}$ is the $B$ at which $g_{Mmax}/2$ is achieved when $R$ is in excess.

Mutualistic H–M community: If Byproduct is harmful for H, then the community is mutualistic: H and M promote the growth of each other. Such a mutualistic community can still be described by Equations (1–5) and (7), but

Equation (6) is replaced with

$$g_H(R) = g_{Hmax}\frac{R}{R + K_{HR}}\exp\left(-\frac{B}{B_0}\right) \tag{8}$$

where larger $B_0$ indicates lower sensitivity, or higher resistance of H to its Byproduct B.

Exploitative H–M community: If M releases an antagonistic byproduct A that inhibits the growth of H, then the interaction is exploitative: H promotes the growth of M, but M inhibits the growth of H. Besides Eqs (1–5) and (7), we then need to add an equation that describes the dynamics of A

$$\frac{d\widetilde{A}}{dt} = r_A g_M(1 - f_P)M$$

where $r_A$ is the amount of A released when M's biomass grows by 1 unit. We can then normalize $\widetilde{A}$ with $r_A$

$$A = \widetilde{A}/r_A$$

so that

$$\frac{dA}{dt} = g_M(1 - f_P)M. \tag{9}$$

We also need to modify the growth rates for H:

$$g_H = g_H(R) = g_{Hmax}\frac{R}{R + K_{HR}}\frac{A_0}{A + A_0} \tag{10}$$

where larger $A_0$ indicates lower sensitivity, or higher resistance of H to M's Antagonistic by product A.

To calculate the community function landscape, species attractor, and Newborn restrictor, all phenotype parameters, except $\overline{f}_P(0)$ take the value from the Bounds column in Table 1. To construct the landscape such as in Fig. 2c, we calculated $P(T)$ for every grid point on a 2D quadrilateral mesh of $10^{-2} \le \phi_M(0) \le 0.99$ and $10^{-2} \le \overline{f}_P(0) \le 0.99$ with a mesh size of $\Delta\phi_M(0) = 10^{-2}$ and $\Delta\overline{f}_P(0) = 10^{-2}$. To construct the landscapes in Fig. 5b(ii) and b(iii), $P(T)$ was similarly calculated on a 2D grid with a finer mesh of $\Delta\phi_M(0) = 5 \times 10^{-3}$ and $\Delta\overline{f}_P(0) = 10^{-4}$.

To calculate the species composition attractor, we integrated Equations (1–5) to obtain $\phi_M(T) - \phi_M(0)$ for each grid point on the 2D mesh of $\phi_M(0)$ and $\overline{f}_P(0)$. The contour of $\phi_M(T) - \phi_M(0) = 0$ is then the species attractor (blue dashed curve in Fig. 2b).

The attractor-induced Newborn restrictor at a given $\overline{f}_P(0)$ is calculated from its definition: if $\phi_M(0)$ of a parent Newborn is on the restrictor, then so is the average $\phi_M(0)$ among its offspring Newborns. Under no spiking, since the average $\phi_M(0)$ among offspring Newborn is the same as $\phi_M(T)$ of their parent Adult, the Newborn restrictor coincides with the species attractor (Fig. 3b and Fig. 5b ii). Under x% H spiking, x% of the biomass in Newborns is replaced with H cells. Thus if the parent Adult's fraction of M biomass is $\phi_M(T)$, the average $\phi_M(0)$ among its offspring Newborns is $(1 - x\%)\phi_M(T)$ under x% H spiking. The Newborn restrictor therefore is the contour of $(1 - x\%)\phi_M(T) - \phi_M(0) = 0$ (teal curve in Fig. 5a ii and b iii, Fig. 2d ii). Compared with the orange restrictor under no spiking, the teal restrictor is shifted down.

**Parameter choices.** Details justifying our parameter choices are given in the Methods section of our previous work[15]. Briefly, our parameter choices are based on experimental measurements of microorganisms (e.g., *S. cerevisiae* and *E. coli*). To ensure the coexistence of H and M, M must grow faster than H for part of the maturation cycle since M has to wait for H's Byproduct at the beginning of a cycle. Because we have assumed M and H to have similar affinities for Resource (Table 1), the maximal growth rate of M ($g_{Mmax}$) must exceed the maximal growth rate of H ($g_{Hmax}$), and M's affinity for Byproduct ($1/K_{MB}$) must be sufficiently large. Moreover, metabolite release and consumption need to be balanced to avoid extreme species ratios. We assume that H and M consume the same amount of Resource per new cell ($c_{RH} = c_{RM}$) since the biomass of various microbes shares similar elemental (e.g., carbon or nitrogen) compositions. We set consumption value so that the input Resource can support a maximum of $10^4$ total biomass. The evolutionary bounds are set, such that evolved H and M could coexist for $f_P < 0.5$, and that Resource was on average not depleted by $T$ to avoid cells entering stationary phase.

In our simulations, we define "mutation rate" as the rate of nonneutral mutations that alter a phenotype. For example in yeast, mutations that increase growth rate by ≥2% occur at a rate of ~$10^{-4}$ per genome per generation (calculated from Fig. 3 of Levy et al.[52]), and mutations that reduce growth rate occur at a rate of $10^{-4} \sim 10^{-3}$ per genome per generation[53,54]. Moreover, mutation rate can be elevated by as much as 100-fold in hypermutators. In our simulations, we assume a high, but biologically feasible, rate of $2 \times 10^{-3}$ phenotype-altering mutations per cell per generation per phenotype to speed up computation. At this rate, an average community would sample ~20 new mutations per phenotype during maturation. When we simulated with a 100-fold lower mutation rate, evolutionary dynamics slowed down, but all of our conclusions still held[15]. Among phenotype-altering mutations, tens of percent create null mutants, as illustrated by experimental studies on protein, viruses, and yeast[53,55,56]. Thus, we assumed that 50% of

phenotype-altering mutations were null (i.e., resulting in zero maximal growth rate, zero affinity for metabolite, or zero $f_P$). Among nonnull mutations, the relative abundances of enhancing versus diminishing mutations are highly variable in different experiments. We based our distribution of mutation effects on experimental studies on *S. cerevisiae* where the fitness effects of thousands of mutations were quantified under various nutrient limitations in an unbiased fashion[57]. The relative fitness changes caused by beneficial (phenotype-enhancing) and deleterious (phenotype-diminishing) mutations can be approximated by a bilateral exponential distribution with means $s_+ = 0.050 \pm 0.002$ and $s_- = 0.067 \pm 0.003$ for the positive and negative halves, respectively.

**Simulating community selection with small population size**. Individual-based stochastic simulation codes used in this work are largely similar to those in our previous work, except for the modification to simulate species spiking. Below, we briefly recapture the flow of the simulation, which can be found in our previous work[15].

Each simulation begins with $n_{tot} = 100$ identical Newborns. The total biomass of each Newborn is $BM_{target}$. In most simulations, $BM_{target} = 100$ consisting of 60 M cells and 40 H cells of biomass 1. Each Newborn is supplied with abundant agriculture waste and a fixed amount of Resource that supports the growth of $10^4$ total biomass. Unless otherwise specified, maturation time is set to $T = 17$ (~6 generations) to avoid Resource depletion (i.e., stationary phase in experiments) and cheater takeover.

Each maturation cycle is divided into time steps of length $\Delta\tau = 0.05$. During each time step, the biomass of each M and H cell grows deterministically according to Equations (4) and (5) (or the corresponding equations for mutualistic and exploitative communities) without the death terms, while the concentration of Resource, Byproduct and Product changes according to Equations (1–3). At the end of each $\Delta\tau$, each M and H cell dies with a probability of $\delta_M\Delta\tau$ and $\delta_H\Delta\tau$, respectively.

Among the survived cells, if a cell's biomass exceeds the threshold of 2, the cell divides into two identical daughter cells. Each daughter cell then mutates with a probability of $P_{mut}$. For a M cell, its $f_P$, $g_{Mmax}$, $K_{MR}$ and $K_{MB}$ can mutate independently. For a H cell, its $g_{Hmax}$ and $K_{HR}$ can mutate independently. Additionally, H's resistance to Byproduct $B_0$ in a mutualistic H–M community and H's resistance to M's antagonistic byproduct $A_0$ in an exploitative H–M community can mutate independently. In our simulations, these biological parameters are inherited upon cell division, and thus we refer to them as phenotypes or genotypes interchangeably. In most simulations of the simple scenario, only $f_P$ of M mutates, while other phenotypes are held at their bounds whose values are shown in the "Bounds" column of Table 1. In simulations where six phenotypes of the commensal H–M community or seven phenotypes of the mutualistic H–M community could be modified by mutations (e.g., Fig. 7 and Supplementary Fig. 12 for the commensal and Supplementary Fig. 20 for the mutualistic community), these phenotypes start from ancestral values shown in the "Ancestral" column of Table 1. In simulations where seven phenotypes of the exploitative H–M community could be modified by mutations (Supplementary Fig. 21), M's $f_P$ and H's sensitivity to M's antagonistic byproduct $A_0$ start from ancestral values shown in the "Ancestral" column of Table 1. The other five growth phenotypes (2 maximal growth rates and 3 affinities to B and R) start from evolutionary bound shown in the "Bounds" column of Table 1. This choice allows us to speed up the simulation, since if we initiate these five growth phenotypes from ancestral values, there is not enough biomass in Adult communities to perform heritability check until after more than 1000 cycles.

If a mutation occurs, it could be a null mutation with a probability of $\frac{1}{2}$. A null mutation reduces $f_P$, $g_{Mmax}$, $g_{Hmax}$, $A_0$, or $B_0$ to zero, while increases $K_{MR}$, $K_{MB}$ and $K_{HR}$ to infinity (equivalent to reducing affinities to zero). If a mutation is not null, it modifies each phenotype by ~5–6% on average. Specifically, each phenotype is multiplied by $(1 + \Delta s)$, where $\Delta s$ is a random variable with a distribution

$$\mu_{\Delta s}(\Delta s) = \begin{cases} \frac{1}{s_+ + s_-(1-\exp(-1/s_-))}\exp(-\Delta s/s_+), & \text{if } \Delta s \geq 0; \\ \frac{1}{s_+ + s_-(1-\exp(-1/s_-))}\exp(\Delta s/s_-), & \text{if } -1 < \Delta s < 0. \end{cases} \quad (11)$$

Here, $s_+ = 0.05$ and $s_- = 0.067$ are the average percentage by which a mutation increases or decreases a phenotype, respectively (for parameter justifications, see our previous work[15]).

At the end of a maturation cycle, the amount of Product P accumulated in the Adult, $P(T)$, is the community function. In some simulations, measurement noise is added to the true $P(T)$ to yield the measured community function. For the simple scenario where only $f_P$ is modified by mutations (e.g., Fig. 6(e–h)), measurement noise is a normal random variable with 0 mean and standard deviation of 100, approximately 10% of the community function of Cycle 1. For the complex scenario where 6 or 7 phenotypes of H and M are modified by mutations (e.g., Fig. 7), measurement noise is a normal random variable with 0 mean and standard deviation of 50. The magnitude of noise is comparable to the ancestral commensal and mutualistic community function, and ~1/4 of the ancestral exploitative community function. Top $n_{chosen}$ Adult communities with the highest measured function are chosen to be reproduced. Sometimes, more than $n_{chosen}$ Adults might be needed to obtain $n_{tot} = 100$ Newborns for the next cycle if there is not enough biomass in $n_{chosen}$ Adults.

Chosen Adults are reproduced into Newborns with different methods. If "cell sorting" is used, then the deviation of a Newborn's total biomass from the target $BM_{target} = 100$ is within 2, and the deviation of a Newborn's species ratio from that of the parent Adult is within 2%. If "pipetting an inoculum of a fixed total biomass" is used, then a Newborn's total biomass is within a deviation of two from the target $BM_{target}$, while its species composition fluctuates stochastically. If "pipetting" is used, then Newborn's total biomass and species composition both fluctuate stochastically. The dilution fold of each Adult is adjusted, so that the average Newborn community's total biomass is $BM_{target}$ over all selection cycles. If a fraction $\varphi_S$ of a Newborn's biomass is to be replaced by M or H cells, each Newborn gets on average a biomass of $BM_{target}(1 - \varphi_S)$ from its parent Adult community and on average a biomass of $BM_{target}\varphi_S$ from M or H-spiking mix. Specifically, suppose that the biomass of an Adult is $BM(T) = M(T) + H(T)$ where $M(T)$ and $H(T)$ are the biomass of M and H at time $T$, respectively. If a fraction $\varphi_S$ of each Newborn's biomass is to be replaced by a spiking mix, this Adult is then reproduced into $n_D$ Newborns, where

$$n_D = \lfloor BM(T)/[BM_{target}(1 - \varphi_S)] \rfloor \quad (12)$$

and $\lfloor x \rfloor$ is the floor (round-down) function. If $n_D$ is larger than $n_{tot}/n_{chosen}$, only $n_{tot}/n_{chosen}$ Newborns are kept. Otherwise, all $n_D$ Newborns are kept and as many additional Adults with the next highest functions are reproduced to obtain $n_{tot}$ Newborns for the next cycle. These Newborns are then topped off with either M or H spiking mixes so that their total biomass is on average $BM_{target} = 100$, as described in the next subsection. Note that the fold of dilution of an Adult is calculated based on biomass, a continuous variable. However, the biomass is composed of individual biomass of discrete cells. During reproduction, integer number of cells is distributed into each Newborn community.

**Simulating species spiking when only M's cost $f_P$ mutates**. In the simple scenario where only $f_P$ of M is modified by mutations, phenotypes of all H cells are the same. Within an Adult community, all H cells also start with identical individual biomass $L_H$, because simulations start with H cells of biomass 1 and because growth is synchronous. To mimic reproducing a chosen Adult through pipetting an inoculum of a fixed total biomass into each Newborn with a $\varphi_S$-H-spiking strategy, H and M cells from the chosen Adult are randomly assigned to a Newborn community, until its total biomass comes closest to $BM_{target}(1 - \varphi_S)$. If $\varphi_S > 0$, the number of H cells supplemented to the Newborn community is the nearest integer to $BM_{target}\varphi_S L_H^{-1}$. Because integer number of cells is assigned to each Newborn, the total biomass might not be exactly $BM_{target}$ but within a small deviation of ~2 biomass units.

To mimic reproducing through pipetting, each M and H cell in an Adult community is assigned a random integer between 1 and dilution factor $n_D$ (Equation (12)). All cells assigned with the same random integer are then dealt to the same Newborn, generating $n_D$ Newborn communities. If $\varphi_S > 0$, the number of H cells supplemented into each Newborn is a random number drawn from a Poisson distribution of a mean of $BM_{target}\varphi_S L_H^{-1}$.

To mimic reproducing through cell sorting, each Newborn receives a biomass of $BM_{target}(1 - \varphi_S)$ from its parent Adult. Suppose that the fraction of M biomass in the parent Adult is $\phi_M(T)$, then M cells from the parent Adult are randomly assigned to the Newborn, until the total biomass of M comes closest to $BM_{target}\phi_M(T)(1 - \varphi_S)$ without exceeding it. H cells with a total biomass of $BM_{target}(1 - \phi_M(T))(1 - \varphi_S)$ are assigned similarly. If $\varphi_S > 0$, the number of H cells supplemented to the Newborn community is the nearest integer to $BM_{target}\varphi_S L_H^{-1}$ where $L_H$ is the biomass of individual H cell in the parent Adult. Because each of M and H cells had a length between 1 and 2, the actual biomass of M and H assigned to a Newborn could vary from the target by up to 2 biomass units. Consequently, deviations of $BM(0)$ from $BM_{target}$ and of $\phi_M(0)$ from parent Adult's $\phi_M(T)$ are only a few percent.

**Simulating species spiking when both H and M cells evolve**. In the more complex scenario, both H and M evolve. We thus need to spike with evolved H and M clones. Additionally, Newborns are spiked with H or M clones from their own lineage as demonstrated in Supplementary Fig. 11a. Below, we describe the simulation code for the experimental procedure (Supplementary Fig. 11a) we simulated.

In all simulations where 6 or 7 phenotypes are modified by mutations, chosen Adults are reproduced through pipetting in a similar fashion as described above. After Newborns are reproduced from a chosen Adult in Cycle $C - 1$, a preset number of H or M cells are randomly picked from the remaining of this Adult to form H or M-spiking mix for Cycle $C$. At the end of Cycle $C$, we choose 10 Adults with the highest functions. Assuming that each chosen Adult is reproduced through pipetting with $\varphi_S$-H-spiking strategy, a Newborn receives on average a biomass of $BM_{target}(1 - \varphi_S)$ from its parent Adult community and on average a biomass of $BM_{target}\varphi_S$ from H spiking mix generated at the end of Cycle $C - 1$. Since each chosen Adult usually gives rise to 10 Newborns, the number of cells distributed from the chosen Adult to each Newborn is drawn from a multinomial distribution. Specifically, denote the integer random numbers of cells that would be assigned to 10 Newborns to be $\{x_1, x_2, ..., x_{10}\}$. If the chosen Adult has a total

biomass of $BM(T)$ composed of $I_M$ M cells and $I_H$ H cells (both $I_M$ and $I_H$ are integers), the probability that $\{x_1, x_2, ..., x_{10}\}$ cells are assigned to 10 Newborns, respectively, and $x_{11}$ cells remain, is

$$\Pr(\{x_1, x_2, ..., x_{10}, x_{11}\}) = \frac{(I_H + I_M)!}{x_1! \cdots x_{10}! x_{11}!} p_0^{\ x_1 + \cdots + x_{10}} p_{11}^{\ x_{11}}.$$

Here, $p_0 = BM_{\text{target}}(1 - \varphi_S)/BM(T)$ is the probability that a cell is assigned to one of 10 Newborns, $p_{11} = 1 - 10 p_0$ is the probability that a cell is not assigned to Newborns. Thus, $x_{11} = I_H + I_M - \sum_{i=1}^{10} x_i$ is the number of cells remaining after reproduction, from which H and M cells are randomly picked to generate the spiking mix for Cycle $C + 1$.

Suppose that the current spiking strategy is $\varphi_S$-H, then these 10 Newborns are spiked with H-spiking mix generated in Cycle $C - 1$. An average of $BM_{\text{target}}\varphi_S$ of H biomass is spiked into each Newborn so that the total biomass of Newborns is on average $BM_{\text{target}}$. Suppose that five H cells from the parent Adult's lineage are randomly picked at the end of Cycle $C - 1$, and that they have biomass $\{L_{H1}, L_{H2}, L_{H3}, L_{H4}, L_{H5}\}$, respectively. The total number of H cells assigned to each Newborn, $x_H$, is then randomly drawn from a Poisson distribution with a mean of $BM_{\text{target}}\varphi_S/\overline{L}_H$, where $\overline{L}_H = \frac{1}{5}\sum_{j=1}^{5} L_{Hj}$ is the average biomass of the five H cells. Each spiked H cell has an equal chance of being one of the five cells.

**Updating spiking percentage based on heritability checks**. When the community function landscape is unknown, we can estimate heritability of community function under different spiking percentages through parent–offspring regression. In most simulations (e.g., Fig. 7), heritability evaluation is carried out about every 100 cycles ("periodic heritability check"). In the simulations demonstrated in Supplementary Fig. 17, the average improvement rate in community function is estimated from the chosen Adults over the last 50 cycles. Heritability evaluation is carried out when this average improvement rate becomes negative ("adaptive heritability check"). For both periodic and adaptive checks, heritability evaluation can be postponed until within-community selection improves cell growth sufficiently to provide sufficient biomass for heritability check.

During one round of heritability evaluation, heritability of community function is estimated through parent–offspring community function regression under all candidate spiking strategies (Supplementary Fig. 11b). The current spiking strategy is updated if an alternative spiking strategy confers significantly higher community function heritability.

To evaluate heritability under one spiking strategy, up to 100 Newborn communities are generated under this spiking strategy. After these mature into Adults, their functions are the parent functions. Each Adult parent then gives rise to six Newborn offspring under the same spiking strategy. When the six Newborn offspring mature into Adults, the median of their functions is the average offspring function. When offspring functions are plotted against their parent functions, the slope of the least-squares linear regression (green dashed line in Supplementary Fig. 11b) quantifies the heritability of community function. Heritability of a community function is thus similar to heritability of an individual trait, except that we use median instead of mean of offspring functions, because median is less sensitive to outliers. The 95% confidence interval of heritability is then estimated by nonparametric bootstrap[58,59]. More specifically, first, 100 pairs of parent–offspring community functions are resampled with replacement. Second, heritability is calculated with the resampled data. Third, 1000 heritabilities are calculated from 1000 independent resamplings, from which the 95% confidence interval is estimated from the 5th and 95th percentile.

An alternative spiking strategy is considered significantly more advantageous than the current spiking strategy if heritability of the alternative spiking strategy is higher than the right endpoint of the 95% confidence interval of the heritability of the current spiking strategy. If more than one alternative spiking strategies are more advantageous, the one with the highest heritability is implemented to replace the current strategy. Similarly, an alternative spiking strategy is considered more disadvantageous if heritability of the alternative spiking strategy is lower than the left endpoint of the 95% confidence interval of the heritability of the current spiking strategy. When implementing random spiking strategy, the current spiking strategy is updated with a strategy randomly picked from candidate spiking strategies.

**Simulating community selection with large population size**. When the population size of each community is scaled up by 10 or 100 times (Supplementary Figs. 2 and 18b), the simulation codes described above become inefficient. Instead of tracking the biomass and phenotype of each cell in a large population, we divide the cells into categories and track the number of cells from different categories, where a category is defined by a unique combination of cell biomass and phenotype ranges. In our simulations, the biomass of each cell ranges between 1 and 2, $f_P$ of each M cell ranges between 0 and 1. Since H cells do not mutate, H cells are divided into 100 categories. H cells that belong to category $i$ have a biomass between $[1 + (i - 1) \times \Delta L, 1 + i \times \Delta L]$ where $\Delta L = 10^{-2}$. Since only $f_P$ of M cells are modified by mutations, M cells are divided into $100 \times 10^5$ categories. M cells that belong to category $(i, j)$ have a biomass between $[1 + (i - 1) \times \Delta L, 1 + i \times \Delta L]$ and $f_P$ between $[(j - 1) \times \Delta f_P, j \times \Delta f_P]$ where $\Delta f_P = 10^{-5}$. Every time $f_P$ of a M cell is modified by mutations, this cell jumps from the current category to a new category determined by its new $f_P$ value.

Similar to simulations with small population sizes, each selection cycle starts with $n_{\text{tot}} = 100$ Newborn communities. Maturation time $T$ is divided into time steps of length $\Delta\tau = 0.05$. Over each time step, the growth in cell biomass and the changes in metabolites are simulated in a similar fashion as described above. At the end of each time step, the number of cells to die or to mutate in each category is drawn from a binomial distribution. If $f_P$ of a M cell is modified by mutation, the mutation effect is drawn from the same distribution as described above: $\frac{1}{2}$ of mutations reduce $f_P$ to 0 and the other $\frac{1}{2}$ is randomly drawn from the distribution in Equation (11).

At the end of a maturation cycle, top 10 Adults with the highest functions are chosen. Each then reproduces 10 Newborns via pipetting for the next cycle. The fold of dilution is similarly adjusted, so that the average of Newborn total biomass is $BM_{\text{target}}$ over all selection cycles. From each category of a chosen Adult, the number of cells assigned to a Newborn community is randomly drawn from a multinomial distribution.

**Reporting summary**. Further information on research design is available in the Nature Research Reporting Summary linked to this article.

## Data availability
The data that support the findings of this study are available in https://github.com/shougroup/Xie_Shou_2021_SteeringEcoEvoDynamics/.

## Code availability
All codes used in this study are available in https://github.com/shougroup/Xie_Shou_2021_SteeringEcoEvoDynamics/.

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

## Acknowledgements

We are grateful for the suggestions from the other Shou lab members: Caroline Cannistra, David Skelding, Sonal, and Alex Yuan. This work is supported by US National Institutes of Health (R01GM124128), US National Science Foundation (#1917258), UK Academy of Medical Sciences Professorship, and UK Royal Society Wolfson Fellowship.

## Author contributions

L.X. designed the study, performed the simulations, analyzed the data, and wrote the paper. W.S. designed the study, analyzed the data, and wrote the paper.

## Competing interests

The authors declare no competing interest.
