## [Peer Review File · Nature Communications]

Peer Review Comments, initial response to Manuscript:

Reviewer #4 (Remarks to the Author):

My overall impression is that this is an important article, diving into the heart of the question of how directed evolution can be best used to generate microbial consortia with desired properties. The approach by Xie and Shou, based on their prior work, is conceptually and mathematically simple, but extremely rich and far from obvious in its capacity to explore different scenarios. I will admit that this was not an easy read for me, partially due to the fact that I am not very familiar with some of the prior work in evolutionary genetics that the article refers to. At the same time, the key results are explained quite clearly, and the analyses performed are very meticulous. I find it quite exciting that, despite its abstract and theoretical nature, the article provides precise guidelines that could be used in experimental testing and implementations.

The reviewers that provided prior comments seemed overall similarly positive about the value of this article and raised a number of interesting questions, which the authors addressed very thoroughly through new analyses and major revisions to the text. My impression is the authors took all comments very seriously and put a lot of effort into appropriately addressing the criticism. In fact, I feel that the debate that emerges from the reviewers' comments and authors' responses is in itself quite interesting, and demonstrates the high value of this article, suggesting that researchers from the evolutionary and microbial ecology fields will be very intrigued by this work. My recommendation would be to accept this paper for publication.

Reviewer #5 (Remarks to the Author):

Summary:

In this paper, the authors explore: 1) how non-heritable determinants can reduce community-level heritability, lowering the effectiveness of selection, 2) how ecological dynamics can reduce heritability and "trap" communities in certain parts of the functional landscape, and 3) propose a way forward by periodically introducing disturbances (moving communities away from their "trap") that increase heritability.

Comments:

Overall the paper was very clear, and an important contribution to the field. After reading the response to reviewers, I thought that the authors have done a good job increasing clarity and streamlining the argument. Nevertheless, I still felt like some important points were hard to understand immediately, and that maybe some added clarity could strengthen the paper.

In particular, the difference between "Newborn restrictor" and attractor is an important one, but it is one that can be confusing. I think it could be helpful to remind readers that communities are "trapped" by an ecological attractor (as adults) and that the attractor restricts the composition of newborns. I think Figure S6A is really clear and I wonder if there is a way to bring it into the main text.

Finally, it is not clear in the main text how do you get the right manipulation to increase heritability? After reading the supplementary material, it is clear that the answer is doing a series of

perturbations and evaluating the effect of each one on heritability. Given that these heritability checks are the main proposal to improve selection, I would consider explaining this in the main text.

August 29, 2021

Our responses to Reviewers' comments are presented below.

Reviewer #4 (Remarks to the Author):

My overall impression is that this is an important article, diving into the heart of the question of how directed evolution can be best used to generate microbial consortia with desired properties. The approach by Xie and Shou, based on their prior work, is conceptually and mathematically simple, but extremely rich and far from obvious in its capacity to explore different scenarios. I will admit that this was not an easy read for me, partially due to the fact that I am not very familiar with some of the prior work in evolutionary genetics that the article refers to. At the same time, the key results are explained quite clearly, and the analyses performed are very meticulous. I find it quite exciting that, despite its abstract and theoretical nature, the article provides precise guidelines that could be used in experimental testing and implementations. The reviewers that provided prior comments seemed overall similarly positive about the value of this article and raised a number of interesting questions, which the authors addressed very thoroughly through new analyses and major revisions to the text. My impression is the authors took all comments very seriously and put a lot of effort into appropriately addressing the criticism. In fact, I feel that the debate that emerges from the reviewers' comments and authors' responses is in itself quite interesting, and demonstrates the high value of this article, suggesting that researchers from the evolutionary and microbial ecology fields will be very intrigued by this work. My recommendation would be to accept this paper for publication.

We thank Reviewer 4 for patiently reading through not only our manuscript, but also a very lengthy rebuttal. We have prepared two video walkthroughs, one by Li Xie and one by Wenying Shou. We hope that these walkthroughs will help readers better understand our work.

Reviewer #5 (Remarks to the Author):

Summary:

In this paper, the authors explore: 1) how non-heritable determinants can reduce community-level heritability, lowering the effectiveness of selection, 2) how ecological dynamics can reduce heritability and "trap" communities in certain parts of the functional landscape, and 3) propose a way forward by periodically introducing disturbances (moving communities away from their "trap") that increase heritability.

Comments:

Overall the paper was very clear, and an important contribution to the field. After reading the response to reviewers, I thought that the authors have done a good job increasing clarity and streamlining the argument. Nevertheless, I still felt like some important points were hard to

understand immediately, and that maybe some added clarity could strengthen the paper.

In particular, the difference between "Newborn restrictor" and attractor is an important one, but it is one that can be confusing. I think it could be helpful to remind readers that communities are "trapped" by an ecological attractor (as adults) and that the attractor restricts the composition of newborns. I think Figure S6A is really clear and I wonder if there is a way to bring it into the main text.

We thank Reviewer 5 for patiently reading through not only our manuscript, but also a very lengthy rebuttal. Thanks for your suggestions. We have now moved Figure S6A to be part of Figure 2d.

Finally, it is not clear in the main text how do you get the right manipulation to increase heritability? After reading the supplementary material, it is clear that the answer is doing a series of perturbations and evaluating the effect of each one on heritability. Given that these heritability checks are the main proposal to improve selection, I would consider explaining this in the main text.

In Figure 5, we chose the spiking strategy so that the contour lines near the Newborn restrictor are largely perpendicular to the axis of the heritable determinant. When the landscape is unknown (Figure 7), you are correct that we simply try a series of perturbations and compare their effects on heritability. We have now changed our text to state:

“Thus, we can try several different perturbation strategies, compare them, and choose the strategy yielding the highest community function heritability.”